# Traumatic Brain Injury in Cameroon: A Prospective Observational Study in a Level I Trauma Centre

**DOI:** 10.3390/medicina59091558

**Published:** 2023-08-28

**Authors:** Franklin Chu Buh, Irene Ule Ngole Sumbele, Andrew I. R. Maas, Mathieu Motah, Jogi V. Pattisapu, Eric Youm, Basil Kum Meh, Firas H. Kobeissy, Kevin W. Wang, Peter J. A. Hutchinson, Germain Sotoing Taiwe

**Affiliations:** 1Department of Animal Biology and Conservation, Faculty of Science, University of Buea, Buea P.O. Box 63, Cameroonmehbasil90@gmail.com (B.K.M.); 2Department of Neurosurgery, Antwerp University Hospital, University of Antwerp, 2000 Edegem, Belgium; andrew.maas@uza.be; 3Department of Surgery, Faculty of Medicine and Pharmaceutical Sciences, University of Douala, Douala P.O. Box 2701, Cameroon; motmath@yahoo.fr; 4Department of Pediatric Neurosurgery, University of Central Florida College of Medicine, 6850 Lake Nona Blvd, Orlando, FL 32827, USA; jpattisapu@ped-neurosurgery.com; 5Holo Healthcare, Nairobi 00400, Kenya; ericprincelds@gmail.com; 6Department of Biochemistry and Molecular Genetics, Faculty of Medicine, American University of Beirut, Riad El-Solh, Beirut P.O. Box 11-0236, Lebanon; 7Center for Neurotrauma, Multiomics & Biomarkers (CNMB), Department of Neurobiology, Neuroscience Institute, Morehouse School of Medicine, 720 Westview Dr SW, Atlanta, GA 30310-1458, USA; kawangwang17@gmail.com; 8Department of Clinical Neuroscience, Cambridge CB2 0QQ, UK; pjah2@cam.ac.uk

**Keywords:** characteristics, traumatic brain injury, outcome, disparities in care, prospective study

## Abstract

*Background and Objective:* About 14 million people will likely suffer a traumatic brain injury (TBI) per year by 2050 in sub-Saharan Africa. Studying TBI characteristics and their relation to outcomes can identify initiatives to improve TBI prevention and care. The objective of this study was to define the features and outcomes of TBI patients seen over a 1-year period in a level-I trauma centre in Cameroon. *Materials and Methods:* Data on demographics, causes, clinical aspects, and discharge status were collected over a period of 12 months. The Glasgow Outcome Scale-Extended (GOSE) and the Quality-of-Life Questionnaire after Brain Injury (QoLIBRI) were used to evaluate outcomes six months after TBI. Comparisons between two categorical variables were done using Pearson’s chi-square test. *Results:* A total of 160 TBI patients participated in the study. The age group 15–45 years was most represented (78%). Males were more affected (90%). A low educational level was seen in 122 (76%) cases. Road traffic incidents (RTI) (85%), assaults (7.5%), and falls (2.5%) were the main causes of TBI, with professional bike riders being frequently involved (27%). Only 15 patients were transported to the hospital by ambulance, and 14 of these were from a referring hospital. CT-imaging was performed in 78% of cases, and intracranial traumatic abnormalities were identified in 64% of cases. Financial constraints (93%) was the main reason for not performing a CT scan. Forty-six (33%) patients were discharged against medical advice (DAMA) due to financial constraints. Mortality was 14% (22/160) and high in patients with severe TBI (46%). DAMA had poor outcomes with QoLIBRI. Only four patients received post-injury physical therapy services. *Conclusions:* TBI in Cameroon mainly results from RTIs and commonly affects young adult males. Lack of pre-hospital care, financial constraints limiting both CT scanning and medical care, and a lack of acute physiotherapy services likely influenced care and outcomes adversely.

## 1. Introduction

Traumatic brain injury (TBI) is considered a global public health challenge, affecting about 69 million individuals annually and is the leading cause of death and disability globally [1,2]. In sub-Saharan Africa (SSA), an estimated 3.2 million people sustain TBI annually, and this number is expected to rise to 14 million by 2050 [3]. Road traffic incidents (RTIs), falls, and assaults are the most common causes of TBI [4,5], with RTIs being the predominant cause in SSA [1,6].

The global TBI burden is increasing, and to a greater extent in SSA, as shown by its fast-rising incidence [7]. In the U.S., about 90,000 people have long-term disabilities due to a TBI [8], but such data are not available from many countries due to limited research. Djientcheu et al. [9] reported TBI outcomes in SSA, with a mortality rate of 77% in severe injuries, 16% in moderate injuries, and 1% in mild cases (again, RTIs were the most common cause). In addition to mortality and morbidity, TBI adversely influences the lives of those affected and their families, often leading to a decreased life expectancy compared with the general population. These injuries often affect lower-income families, and surviving patients incur substantial direct and indirect costs [10,11]. Furthermore, the heterogeneous nature of traumatic brain injuries complicates the accurate assessment of severity and prediction of clinical outcomes [12,13].

Despite the social and economic impact of TBI in Cameroon, data on its characteristics and medium- or long-term outcomes are lacking; only a few studies from SSA report traumatic brain injuries as a major cause of death and disability. Ndoumbe et al. [14] and Motah et al. [15] addressed TBI management and outcomes, but only severe TBI or intracranial haemorrhage were considered, and cases of medium- or long-term outcomes were not included. Furthermore, the study of Ndoumbe et al. [14] was retrospective in design; hence, outcome evaluations were not appropriate for patient characteristics and medium-term outcomes after TBI [16].

Our study seeks to provide patient characteristics and medium-term outcomes with the Glasgow Outcome Scale Extended (GOSE) and Quality of Life after TBI questionnaire (QoLIBRI) from a Level I trauma centre. The study was conducted in Douala, Cameroon to evaluate how TBI patient’s features interact with the outcomes in a LMIC setting.

## 2. Materials and Methods

### 2.1. Study Design

We conducted a prospective cohort study on TBI patients attending a Level I trauma centre at the Laquintinie Hospital in Douala (LHD), Cameroon, from January 2021 to February 2022. Douala is situated on the Atlantic Ocean and is the economic capital of the country, with an estimated population of 4 million [17]. The Laquintinie Hospital manages all types of trauma and is among the few hospitals with technical expertise for TBI care. It has a fully functional neurosurgical unit with four neurosurgeons, a CT scanner, and a 0.5 Tesla MRI.

### 2.2. Study Population

Our study included all patients with TBI received at the LHD within 24 h who were able to consent (on some occasions, from family members). TBI patients with pre-existing neuropsychiatric problems and thieves were excluded, along with those with delayed presentation to LHD (arrived > 24 h after injury, n = 18). We also excluded patients who could not be contacted for the 6-month evaluation (n = 8).

### 2.3. Data Collection

After a definitive diagnosis of TBI, the study purpose was meticulously explained to the patients or their proxies, and consent was obtained to participate in the study. Data collection was based on the NIH-NINDS common data elements and included information such as sociodemographic characteristics (age, sex, profession, marital status, religion, and nationality, monthly income), clinical details (heart rate, blood pressure, temperature, Glasgow Coma Scale (GCS), and neuro-imaging findings), and pre-hospital factors (transport means, time between injury and arrival). Hypothermia was defined as <35 °C, normal between 35 °C and 37.9 °C, and fever if the temperature was >38 °C (CDC, 2019). Hypotension was noted as blood pressure below 90/60 mmHg; values of 130–139/80–89 mmHg were considered stage one hypertension; systolic values > 140 mmHg and/or diastolic values > 90 mmHg were considered stage 2 hypertension [18]. TBI severity was classified according to the GCS: 3–8 for severe, 9–12 for moderate, and 13–15 for mild. We included data on aetiology, injury mechanism, presenting symptoms, hospital length of stay, and outcomes. The GOSE and QoLIBRI were recorded 6 months after hospital discharge by telephone communication with patients and their families (good recovery, mental or physical disability, vegetative state, death). Data collection and reporting were in accordance with the STROBE guidelines (Appendix A).

### 2.4. Outcome Measures

Discharge status was scored using the Disability Rating Scale (DRS) with a total score of 29 (level of disability: 0 none, 1 mild, 2–3 partial, 4–6 moderate, 7–11 moderately severe, 12–16 severe, 17–21 extremely severe, 22–24 vegetative state, 25–29 extreme vegetative state) [19]. The DRS was completed on the day of discharge (including DAMA), and the patient’s welfare was inquired about at 2 and 4 months after discharge to maintain contact and facilitate final evaluations. The 6-month outcome evaluations were performed with the structured interview for GOSE (death, vegetative state, lower or upper severe disability, upper and lower moderate disability, or good recovery) and QoLIBRI (<60 impaired, 60–66 borderline, 67–82 normal, >82 above average). In the majority of the cases, the 6-month outcomes were completed by telephone interviews since patients were from faraway areas (a few were completed in person). To limit the bias of subjectivity on the part of the patients, we interviewed both the patients and their relatives.

### 2.5. Data Management and Analysis

Continuous variables were reported as medians with 25th and 75th percentiles and as means and standard deviations. Categorical variables were described as frequencies and percentages. Comparisons between two categorical variables were done using the chi-square test or Fisher exact test (when the expected frequency was less than 5). *p*-values < 0.05 were considered statistically significant.

### 2.6. Ethical Clearance and Administrative Authorizations

Ethical clearance for the study was obtained from the Institutional Review Board of the Faculty of Health Sciences (IRB-FHS), University of Buea (Reference N° 1238-08), and administrative authorization was obtained from LHD. Informed consent was given by the participants or their family members, and all data collected was kept strictly confidential with physical and electronic barriers.

## 3. Results

### 3.1. Presentation of Sociodemographic and Clinical Features

The study enrolled 160 patients with TBI between January 2021 and February 2022. The median age was 32 (26, 39). Most patients were adolescents/adults aged 15–45 years (78%; 125); 90% of patients were males (144/160). Most participants (76%) had not finished secondary education, and approximately two-thirds were of low economic status (65%). The most common professional group was commercial bike riders (27%; 43/160). Most of the participants who were drivers had no driving licence (81%; 73/90). More than half of the patients (61%; 97) were alcohol consumers or smokers (59%; 95), and pre-existing hypertension was present in 14 (8.8%) patients (Table 1).

### 3.2. Clinical Details of Participants

The two most common clinical manifestations of TBI among participants were headache and loss of consciousness (74%; 119 and 98%; 156), respectively. The median GCS was 12.0 (8.0–14.0), and an eye-opening score of 4, verbal response of 4, and motor response of 6 were the most frequently recorded component scores. There was no pupil reaction in 5 (3.1%) of the cases, and the median heart rate was 88 (78 to 99). About 75% of the victims had a normal temperature (131, or 91%), while 3 (2.1%) had hypothermia. Driving after alcohol consumption was suspected in 33 (21%) of the cases (Table 2).

### 3.3. Injury Details of Participants

The main cause of TBI was RTIs in 136 (85%), and 60% of these involved a motorcyclist. Other causes of TBI admissions at LHD included assault by a blunt instrument (7.5%, 12) and falls (5%, 8) (Table 3).

### 3.4. Pre-Hospital Details of TBI Patients

A total of 95 patients (59%) were secondarily referred from other health facilities, and only 65 (41%) were transported directly from the injury site. The main form of transportation was by non-medical means (91%; 145/160). Only one patient (1/65) was transported from the injury site to the hospital by ambulance. Of those referred from other health structures (n = 95), 14 patients (15%) were transported by ambulances (Table 4).

### 3.5. Injury Characteristics and Management

Isolated TBI occurred in 88% (141/160) of the cases, and only a few cases (12%; 19/160) had TBI and other associated trauma. Mild TBI cases were the most common presentation form (41%; 66/160), followed by moderate (34%; 55/160) and severe (24%; 39/160) TBI. A CT scan was not obtained in 22% of TBI cases, and financial constraint was the main reason. When performed, CT showed traumatic intracranial abnormalities in 64% (77/125) of cases. The two most common types of TBI were cerebral contusion (54%; 65/160) and extradural haemorrhage (49%; 59/160). Skull fractures were seen in 36 (30%) of the cases, and 17 (22% of the 77 victims) required neurosurgical intervention (Table 5).

### 3.6. Discharge Status of Participants

The median hospital stay for non-operated TBI patients was 4.0 days (2.0–6.8), while those who underwent neurosurgery stayed for 7 days (4–11). Twenty-one patients died (13%), of whom 18 had sustained a severe TBI. The median time between injury and death was 24 h. A total of 121 (79%) patients were discharged home, and only 4 (3%) went to rehabilitation services (physiotherapy). Forty-six patients (33%) were discharged against medical advice (DAMA), with financial constraints as the main reason. In survivors, the median disability rating score recorded was 4 (1–10), with 18% of patients (25/139) scoring a good recovery (DRS = 0), as seen in Table 6.

### 3.7. Six Months Outcome after Hospital Discharge

At 6-months, 25% of patients (38/152) had an unfavourable outcome (GOSE 1–4), and 75% (114/152) had a favourable outcome (GOSE 5–8). “One DAMA patient had died since discharge, increasing the overall mortality rate to 14% (22). The GOSE assessment was missing for eight patients. The QoLIBRI identified 36 patients as impaired (28%; 36/130), as shown in Table 7a. Differences in outcome between surviving patients discharged against medical advice (DAMA) versus those completing hospital treatment were explored; no clear difference (*p* = 0.6) was found in the unfavourable outcomes of 15% (6/40) and 12% (11/90), respectively, but a significant difference (*p* = 0.019) was found in the number of patients with impaired scores on the QoLIBRI of 39% (16/41) vs. 22% (20/89) as shown in Table 7b.

### 3.8. Comparison of Six-Month TBI Outcomes and Time Difference between Injury and Arrival, Type of Referral, and TBI Severity

There was no statistically significant difference in outcomes comparing TBI severity and arrival at the hospital (0.847 and 0.577, DRS, QoLIBRI, and GOSE, respectively) (Appendix A). Likewise, we found no difference between direct or indirect referrals and discharge outcomes in the DRS (*p* = 0.061), the 6-month outcomes using the GOSE (*p* = 0.067), or the quality of life after brain injury score (*p* = 0.8) (Appendix A). As expected, a statistically significant correlation was seen with the injury severity and outcomes (discharge and 6-months) for DRS, GOSE, and QoLIBRI (Appendix A).

## 4. Discussion

This study presented the characteristics of TBI and outcomes at a level one trauma centre in Douala, Cameroon, with the aim of identifying measures to improve TBI prevention and care.

### 4.1. Demographic and Injury Characteristics

We enrolled 160 TBI cases, of whom the majority (78%) were aged 15–45 years and predominantly male (90%), consistent with other reports from LMICs [20,21]. Most of the participants (76%, 122/160) had not finished secondary education. As noted by Ashkan et al. [22] and Amram et al. [23], low educational levels constitute one of the specific risk factors for RTIs and assaults and are greatly understudied in low-resource settings. RTIs were the main cause of TBI (85%) and often involved commercial bike riders, which is in line with a 5-year retrospective study conducted in Cameroon by Buh et al. [24]. In Cameroon, many youths are involved in hazardous occupations like commercial bike riding as better jobs are difficult to obtain. In addition, recent socio-political insecurity in the two English-speaking regions (the Anglophone crisis) has led to a rural exodus of youths into Douala. These young citizens are frequently involved in commercial bike riding, often without helmets, and seldom have a driving license, and are hence ignorant of the highway codes and traffic laws [25]. In contrast to LMICs, falls are the main cause of TBI in developed countries [26,27]. This difference in the causes observed can be explained by the fact that HICs have better road infrastructure and the implementation of safety road practises is stricter than in most LMICs, including Cameroon.

Many of the TBI patients involved in the RTIs (81%, 73/90) were drivers without a license. This high percentage of motor vehicle and motorcycle drivers without driving licences is in line with the Cameroon government’s 2019 statistics, where 70% of RTIs were reported to originate from human causes, including a lack of driving licences or illegal driving licences [28]. Furthermore, the use of helmets and seat belts is not a common practise in Cameroon and SSA [29]. The low socioeconomic status of TBI patients could explain the relatively high DAMA cases, as suggested by Amram et al. [23]. Moreover, the majority of the citizens are not insured, hence not able to meet the financial costs of TBI care.

More than half of the TBI victims were alcohol consumers (61%), which corroborates a study in Cameroon that reported high alcohol consumption rates of 41% of men and 25.8% of women [30]. Our result, however, differs from the study of Wakabayashi et al. [31] in Thailand, where only 11.6% of TBI cases consumed alcohol. This low alcohol consumption in Thailand may be explained by religious restrictions compared to Cameroon, where the majority are Christians who, in most cases, are not restricted from alcohol consumption. Alcohol consumption before driving, a reported risk factor for RTIs, was suspected in 21% of cases, which is frequently reported in the literature [32,33]. Hypotension, often associated with negative outcomes after TBI, was registered in 19 (12%) cases, twice the rate reported by Landes et al. [33] in Ethiopia.

### 4.2. Pre-Hospital Care and Post-Acute Care

The results of our study are highly illustrative of the specific challenges and disparities in care in low-resource settings. The main transportation means of TBI cases were through non-medical means (91%), similar to reports by Motah et al. [15] and Buh et al. [24]. Only one patient was transferred from the scene of the accident to the study centre by ambulance. Of patients secondarily referred, only 14 (15%) inter-hospital transports were by ambulance; the others were by taxi or private vehicle. These observations highlight the deficiency of emergency transport systems in LMICS, like Cameroon.

The median length of stay (LOS) for non-operated TBI victims was 4.0 days (2.0, 6.8), which is in line with the study of Tesfay et al. [34] in Ethiopia and Buh et al. [24] in a 5-year retrospective in a Level 1 trauma centre in Douala, Cameroon. Only four (3%) were recommended rehabilitation services (physiotherapy). Unfortunately, acute physiotherapy services were not recommended for hospitalised patients with disabilities, as there are no local standards for physiotherapy referrals for TBI patients. Such therapy is recommended to enhance recovery [35,36,37,38]. The lack of appropriate post-acute care constitutes a major problem in the care of TBI patients, especially in low-income settings, which may greatly impact the outcome of TBI patients. Acute physiotherapy services in TBI can deliver better functional outcomes [38] in SSA, and implementation of these should be high on the policy agenda.

### 4.3. Disparities in Care due to Financial Constraints

Substantial disparities in care due to deficiencies in the health care system and the financial constraints of patients (or their relatives) were identified, as illustrated in Figure 1. CT scanning on presentation was clinically indicated but not performed in 28 out of 160 patients, of whom 35% (10/28) had moderate or severe TBI. In 26 of these (93%), the reason was the inability of patients or their relatives to pay the costs of a CT scan (75,000 FCFA = 122 USD). Forty-six patients (33%) were discharged against medical advice, with financial constraints as the main reason (100%). This rate of DAMA is much higher than the reported rates in high-income countries of 2.8%, 1.9%, and 1.8% by Kim et al. [39], Guise et al. [40], and Marcoux et al. [41]. DAMA cases are disproportionately high in low-income settings like Cameroon. Furthermore, the observed rate of 33% is likely an underestimation of the true financial constraint on treatment after TBI in low-resource settings, as a large part of the population lives in poverty. Some TBI patients without financial means remained in the hospital as their family members tried to raise the necessary funds. Two patients registered in this study received treatment and were later not permitted to leave the hospital premises until they settled their debts. This situation likely caused psychological distress and negatively impacted their recovery, as suggested in the literature [42,43,44]. Findings from this study revealed that survivors of DAMA cases were more likely to have disabilities and a poorer quality of life six months after injury than survivors discharged regularly (*p* = 0.019 for QoLIBRI). No clear difference was found on the GOSE. However, there was a trend towards lower GOSE in DAMA cases, and the absence of statistical significance is likely due to the relatively low sample size. Unfortunately, the negative consequences of financial constraints, such as DAMA, remain largely unreported in LMICs where the burden of TBI is significant.

### 4.4. Outcome

Mortality occurred in 22 patients (14%), consistent with the 10–24% mortality reported by Pelieu et al. [45] in Switzerland, Tesfay et al. [34] in Ethiopia, and El-Menyar et al. [21] in the Middle East. The median time between injury and death was 1 day, much lower than reported in the study of Gao et al. [46], where mortality occurred within 14 days. This difference can likely be explained by the exclusion of cases that died at the emergency service in the study by Gao et al. [46] and also by the higher healthcare standards in China compared to Cameroon. The mortality rate of severe TBI victims in this study (46%, 18/39) is disproportionately high compared with high-income countries. Lu et al. [47] reported a mortality rate of 27% for patients with severe TBI in 1996 in HICs, and the European CENTER-TBI study (2014–2017) reported a mortality rate of 27.8% (Maas, personal communication). This higher death rate with severe TBI in LMICs could be explained by little or no pre-hospital care, limited resources, and financial constraints faced by patients to obtain the necessary health care. Another factor that would have influenced adverse mortality in severe TBI in this study and that may be overlooked in low-income settings is the lack of intracranial pressure monitoring, as previously reported by Buh et al. [48].

The 6-month GOSE outcomes showed that 44 (29%) patients had an upper-good recovery, 21% had an upper-moderate disability, and 12 (8%) had an upper-severe disability, consistent with reports from Paris, France [49] and Tanzania [6]. Ndoumbe et al. [14] from Cameroon noted that only 12.6% of their patients fully recovered and 55% of cases had permanent disability; however, only severe TBI patients were considered in this study. Using the QoLIBRI, 36 (28%) of patients were impaired, while 46 (36%) had above-average scores, consistent with reports by Born et al. [50] in Switzerland.

### 4.5. Strengths and Limitations

This is one of the very few studies designed to present the characteristics of TBI patients and their interaction with the outcome in Cameroon. A major strength of this study is its prospective design with the collection of 6-month outcome data and the fact that we outline and discuss poor standards of prehospital and post-acute care and describe how DAMA due to financial constraints adversely affects management and outcome. The results can inform healthcare policies to improve prevention and develop strategies to achieve the best care services, aimed at improving outcomes for patients with TBI in Cameroon and SSA. We hope to increase the sample size and collect data such as O_2_ saturation, duration before CT, and treatment. Another limitation was the difficulty in following up with patients at the hospital; TBI patients were sometimes admitted to other services/wards, making complete and timely data collection challenging. A few cases (n = 8) were also missed at the 6-month evaluation due to incomplete or inaccurate contact information. Despite these limitations, our results are pertinent, offering insights into the TBI situation in an LMIC. The challenges described in Cameroon are illustrative of the problems that governments and health systems face in low-resource settings.

## 5. Conclusions

Traumatic brain injury in Cameroon commonly occurs in young males of low educational (76%) and socioeconomic status and often involves commercial bike riders. The main cause of TBI in Cameroon is road traffic accidents. Mortality in patients with severe TBI is disproportionately high compared to high-income countries. We report deficiencies in prehospital and postacute care, which likely negatively impact care and outcome when added to the already poor health resources and infrastructure in resource-limited settings. Substantial disparities in care caused by financial constraints were identified as having a clear adverse effect on outcomes. Implementation of universal health insurance may be expected to improve hospital care and outcomes and reduce the number of DAMA cases. Road traffic safety and the prevention of RTIs should be high on the policy agenda. In terms of care provision, improvement of pre-hospital care and development of post-acute care facilities should be the highest priority.

## Figures and Tables

**Figure 1 medicina-59-01558-f001:**
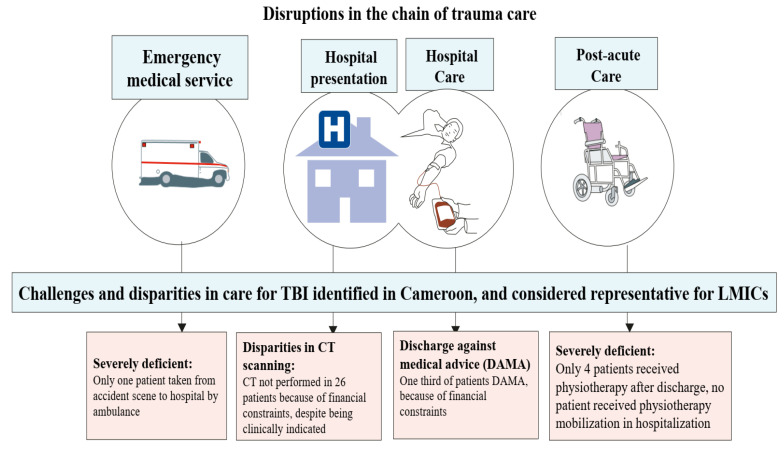
Disruptions in the chain of trauma care. TBI: Traumatic Brain Injury, LMICs: Low-Middle-Income Countries, CT: Computed Tomography, DAMA: Discharged Against Medical Advice, H: Hospital.

**Table 1 medicina-59-01558-t001:** Sociodemographic characteristics.

Characteristic	N = 160
Age, Median (IQR)	32 (IQR26, 39)
[<15]	6 (3.8%)
[15–45]	125 (78%)
[46–65]	22 (14%)
[>65]	7 (4.4%)
Gender	
Female	16 (10%)
Male	144 (90%)
Education	
Graduated	19 (12%)
Illiterate	9 (5.6%)
Matriculated	7 (4.4%)
Post-graduate	1 (0.6%)
Primary	40 (25%)
Secondary	82 (51%)
Not Known	2 (1.2%)
Profession	
Employee in service	30 (19%)
Manual workers	24 (15%)
Bike riders	43 (27%)
Student	16 (10%)
Unemployed	21 (13%)
Others	26 (16%)
Marital status	
Married	67 (42%)
Not applicable	6 (3.8%)
Single	87 (54.4%)
Religion	
Christian	122 (80.3%)
Muslim	28 (18%)
Other	2 (1.3%)
Missing	8
Driving license?	
Yes	11 (12.2%)
No license	73 (81%)
Not applicable	70 (44%)
Unknown	6 (6.7%)
Income rate per month in Central African FCFA (USD)	N = 130
<50,000 (<$78)	26 (20%)
50,001–150,000 ($78–$234)	84 (65%)
150,001–200,000 ($235–$312)	14 (11%)
200,001–300,000 ($313–$468)	2 (2%)
>300,000 (>$468)	4 (3%)
Not applicable	30 (19%)
Medico-social history	
Diabetes	2 (1.2%)
Hypertension	14 (8.8%)
Smoking	22 (14%)
Alcohol	97 (61%)

Applicable refers to TBI patients who were drivers of either a motorcycle or vehicle at the time of the injury. Unapplicable refers to TBI patients who were not driving at the time they sustained the injury.

**Table 2 medicina-59-01558-t002:** Clinical details of TBI patients.

Characteristic	N = 160
Clinical details	
Loss of consciousness	152 (95%)
Vomiting	55 (34%)
Nausea	21 (13%)
Ear bleed	20 (12%)
Nasal bleed	43 (27%)
Headache	103 (64%)
Seizure	7 (4.4%)
Agitation	43 (27%)
Median heart rate	88 (78, 99)
Blood pressure	N = 160
Elevated	15 (9.4%)
Hypertension	62 (39%)
Hypotension	19 (12%)
Normal	64 (40%)
temperature	N = 144
Hypothermia	3 (2.1%)
Normal	131 (91%)
Elevated	10 (6.9%)
Missing	16
Pupil reactivity	
Both pupils reactive	145 (91%)
No pupil reactive	5 (3.1%)
One pupil reactive	10 (6.2%)
Median Glasgow Coma Scale	12.0 (8.0, 14.0)
Eye opening	
1	29 (18%)
2	23 (14%)
3	49 (31%)
4	59 (37%)
Verbal response	
1	34 (21%)
2	18 (11%)
3	25 (16%)
4	46 (29%)
5	37 (23%)
Motor response	
1	4 (2.5%)
2	4 (2.5%)
3	8 (5.0%)
4	22 (14%)
5	45 (28%)
6	77 (48%)
Influence of alcohol	
None	108 (68%)
Suspected	33 (21%)
Unknown	19 (12%)

**Table 3 medicina-59-01558-t003:** Injury details of TBI patients.

Characteristic	N = 160
Cause of injury	160
Assault/violence	12 (7.5%)
Fall	8 (5.0%)
Road traffic incident	136 (85%)
Other Causes	4 (2.5%)
Road traffic collision type	136
Pedestrian	31 (23%)
RTC car (passenger)	7 (5.2%)
RTC cyclist	1 (0.7%)
RTC driver	2 (1.5%)
RTC Motorcyclist	81 (60%)
RTC Motorcyclist (passenger)	11 (8.1%)
RTC other type of vehicle (driver)	1 (0.7%)
Other	1 (0.1%)
Fall type	N = 8
Fall from height	6 (75%)
Fall standing height	2 (25%)
Assault type	N = 12
Assault (without a weapon)	3 (25%)
Assault blunt instrument	5 (42%)
Assault knife/Machete	4 (33%)
Domestic violence	N = 3
Assault (without a weapon)	2 (67%)
Assault blunt instrument	1 (33%)
Other causes	N = 3
Hit by falling object	1 (33%)
Industrial accident	2 (67%)
Mechanism of injury	160
Bicycle accident	1 (0.6%)
Fall from a higher lever	6 (3.4%)
Fall from the same level	5 (3.1%)
Hit by a blunt object	6 (3.4%)
Knife/machete	4 (2.5%)
Motor vehicle accident	23 (14.4%)
Motorcycle accident	111 (69.4%)
Other	4 (2.5%)
If other, specify	
Blow	1 (25%)
hit by falling object	2 (50%)
occupational accident	1 (25%)

RTC: Road Traffic Collision.

**Table 4 medicina-59-01558-t004:** Pre-hospital details of study participants.

Characteristic	N = 160
Referral details	
Direct referral	65 (41%)
Indirect referral	95 (59%)
If direct referral, Means of transport	
Ambulance	1 (1.6%)
Moto taxi	7 (11%)
Private vehicle	10 (16%)
Taxi	45 (67%)
Other	2 (3.3%)
Transport from other health centre to trauma centre	95
Ambulance	14 (15%)
Moto taxi	5 (5%)
Private vehicle	15 (16%)
Taxi	60 (63%)
Other	1 (1.0%)
General means of transport	N = 160
Medical means	15 (9%)
Non-medical means	145 (91%)
If Ambulance,	14
Physician present?	N = 14
No	12 (85.7%)
Yes	2 (14.3%)
Vital monitoring	12 (85.7%)
Fluid administration	0 (0%)
Airway protection	7 (50%)
Drug administration	0 (00%)
Transport position	160
Side lying	28 (18%)
Sitting	44 (28%)
Supine	88 (55%)
Time difference between injury and arrival at the referral hospital	
<1 h	39 (24%)
1–4 h	71 (44%)
4.1–11 h	30 (19%)
11–24	20 (12%)

**Table 5 medicina-59-01558-t005:** Characterization of injury, neuro-imaging, and management.

Characteristic	N = 160
Poly-trauma	19 (12%)
Isolated TBI	141 (88%)
Classification of TBI	N = 160
Mild	66 (41%)
Moderate	55 (34%)
Severe	39 (24%)
CT not done	35 (22%)
CT Scan asked, not done	28 (80%)
No CT scan recommended	7 (20%)
Reason why CT Scan not done	28
No finance	26 (93%)
Other	10 (36%)
If other, specify	10
moved to other hospital	3 (22%)
not asked	7 (78%)
Complementary exams done to characterize injury	N = 160
CT Scan	125 (78%)
If scan or MRI, traumatic abnormalities present	77 (64%)
Fracture	N = 125
Yes	36 (30%)
Linear skull fracture	20 (56%)
Depressed fracture	22 (61%)
Basilar skull fracture	19 (53%)
Type of TBI	N = 77
Extradural hematoma	22 (29%)
Acute subdural haemorrhage	18 (23%)
Cerebral contusion	25 (32%)
Cerebral oedema	10 (13%)
Meningeal haemorrhage	8 (10%)
Mass effect pressure	3 (3.9%)
Intracerebral haemorrhage	12 (16%)
Other type of TBI	4 (5.2%)
Neurosurgery	N = 77
Yes	17 (22.1%)
No	60 (77.9%)

CT: Computed Tomography, TBI: Traumatic Brain Injury.

**Table 6 medicina-59-01558-t006:** Discharge status of TBI patients.

Characteristic	N = 160
Total hospital stay (days)	4.0 (2.0, 6.8)
Post Op Stay (days)	7 (4, 11)
Discharge destination	138
Discharge to home	126 (91%)
Discharge to rehabilitation	4 (3.0%)
Other hospital facility	8 (5%)
Discharge against medical advice (DAMA)	46 (33%)
Reason for DAMA	
Financial constraint	46 (100%)
Traditional treatment	8 (17.4%)
Readmission	00
Outcome at discharge	N = 160
Disability rating scale	N = 138, 4 (1, 10)
Recovery status	N = 138
Extreme vegetative state	6 (4.3%)
vegetative state	3 (2.2)
Extremely severe disability	7 (5.0%)
Severe disability	8 (5.8%)
Moderately severe	26 (18.7%)
Moderate	28 (20.1%)
Partial disability	27 (19.4%)
Mild disability	9 (6.5%)
Recovering	25 (18%)
Mortality	21 (13%)
Mortality according to TBI severity	
Mortality in mild TBI	1 (4.8%)
Mortality in moderate TBI	3 (14.3%)
Mortality in severe TBI	17 (80.9%)
Time difference between injury and death (days)	1.00 (1.00, 2.00)

ICU: Intensive Care Unit, TBI: traumatic brain injury, Op: operation.

**Table 7 medicina-59-01558-t007:** (**a**) Outcome at discharge with DRS and 6 months’ outcome with GOSE and QoLIBRI. (**b**) Correlation between discharge against medical advice and 6-month outcome.

(**a**)			
**Characteristic**	**N = 160**		
6-months outcome with GOSE	N = 152		
Death	22 (14.5%)		
GR−	15 (9.9%)		
GR+	44 (28.9%)		
MD−	23 (15.1%)		
MD+	32 (21.1%)		
SD−	4 (2.6%)		
SD+	12 (7.9%)		
Overall			
Death	22 (14%)		
Good recovery	59 (39%)		
Moderate disability	55 (36%)		
Severe disability	16 (11%)		
Missing	8		
6-months outcome with QoLIBRI	N = 130		
Impaired	36 (28%)		
Borderline	14 (11%)		
Normal	34 (26%)		
Above average	46 (35%)		
Mean final score	71.8 (19)		
Missing and death (08 and 22, respectively)	30		
(**b**)			
Characteristic	No N = 114	Yes N = 46	*p*-Value
GOSE	N = 90	N = 40	0.6
Favourable outcome	79 (88%)	34 (85%)	
Unfavourable outcome	11 (12%)	6 (15%)	
(Missing and death), respectively	24 (3 and 21)	6 (5 and 1)	
QoLIBRI	N = 89	N = 41	0.019 *
Impaired	20 (22%)	16 (39%)	
Not impaired (borderline, normal, above average)	69 (78%)	25 (61%)	
Missing	25	05	

GOSE: Glasgow Outcome Scale Extended, QoLIBRI: Quality of Life After Brain Injury, GR−: Lower good recovery, GR+: Upper good recovery, MD−: Lower moderate disability, MD+: Upper moderate disability, SD−: Lower severe disability, SD+: Upper severe disability; GOSE: Glasgow Outcome Scale Extended, QoLIBRI: Quality of Life After Brain Injury. * *p* < 0.05.

## Data Availability

Not applicable.

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
