# Peer review of "Traumatic Brain Injury in Cameroon: A Prospective Observational Study in a Level I Trauma Centre"

_medicina, 2023, doi:10.3390/medicina59091558_

Round 1

Reviewer 1 Report

I would like to congratulate the writers with conducting a well described study on an important topic.

there are a few typos and such, a more serious one is on repeating the sentence on males being overrepresented and secondary school being unfinished in the study population @ the begining of part 4.1 in the Discussion.

The number of fatalities in het severe group is high; I think the report would benefit from mentioning the treatment that these patients received; mainly standardized ICP treatment yes/no, envolving an ICP monitor

English is altright; typos and the above double sentences.

Author Response

I would like to congratulate the writers with conducting a well described study on an important topic.

there are a few typos and such, a more serious one is on repeating the sentence on males being overrepresented and secondary school being unfinished in the study population @ the begining of part 4.1 in the Discussion.

Thanks, Dear Reviewer, the repetitions have been eliminated, this can be verified in lines 219 to 222

The number of fatalities in het severe group is high; I think the report would benefit from mentioning the treatment that these patients received; mainly standardized ICP treatment yes/no, envolving an ICP monitor

Thank you, Dear Reviewer, we have now addressed this worry by discussing further on the lack of ICP monitoring and its possible influence on mortality, this can be found in the discussion, section 4.4, lines 316 to 318. Another factor that would have influenced adversely mortality in severe TBI in this study, that may be overlooked in low income settings, is the lack of intracranial pressure monitoring, as previously reported by Buh et al. [48]”.

Reviewer 2 Report

The manuscript presents a actual study of traumatic brain injury in Cameroon. Overall it is well written. But the organization of tables should be improved for easy reading.

Author Response

The manuscript presents a actual study of traumatic brain injury in Cameroon. Overall it is well written. But the organization of tables should be improved for easy reading.

Thank you, Dear Reviewer, we have reviewed all the tables again, and have readjusted to improve understanding. This concern Tables; 1, 3, 4, 5, 6
